# N501Y mutation of spike protein in SARS-CoV-2 strengthens its binding to receptor ACE2

Fang Tian[1†], Bei Tong[2*†], Liang Sun[3], Shengchao Shi[1], Bin Zheng[1], Zibin Wang[3], Xianchi Dong[3,4*], Peng Zheng[1*]

[1]State Key Laboratory of Coordination Chemistry, Chemistry and Biomedicine Innovation Center (ChemBIC), School of Chemistry and Chemical Engineering, Nanjing University, Nanjing, China; [2]Institute of Botany, Jiangsu Province and Chinese Academy of Sciences, Nanjing, China; [3]State Key Laboratory of Pharmaceutical Biotechnology, School of Life Sciences, Nanjing University, Nanjing, China; [4]Engineering Research Center of Protein and Peptide Medicine, Ministry of Education, Nanjing, China

*For correspondence:
beitong@cnbg.net (BT);
xianchidong@nju.edu.cn (XD);
pengz@nju.edu.cn (PZ)

[†] These authors contributed equally to this work

Competing interest: The authors declare that no competing interests exist.

**Abstract** SARS-CoV-2 has been spreading around the world for the past year. Recently, several variants such as B.1.1.7 (alpha), B.1.351 (beta), and P.1 (gamma), which share a key mutation N501Y on the receptor-binding domain (RBD), appear to be more infectious to humans. To understand the underlying mechanism, we used a cell surface-binding assay, a kinetics study, a single-molecule technique, and a computational method to investigate the interaction between these RBD (mutations) and ACE2. Remarkably, RBD with the N501Y mutation exhibited a considerably stronger interaction, with a faster association rate and a slower dissociation rate. Atomic force microscopy (AFM)-based single-molecule force microscopy (SMFS) consistently quantified the interaction strength of RBD with the mutation as having increased binding probability and requiring increased unbinding force. Molecular dynamics simulations of RBD–ACE2 complexes indicated that the N501Y mutation introduced additional π-π and π-cation interactions that could explain the changes observed by force microscopy. Taken together, these results suggest that the reinforced RBD–ACE2 interaction that results from the N501Y mutation in the RBD should play an essential role in the higher rate of transmission of SARS-CoV-2 variants, and that future mutations in the RBD of the virus should be under surveillance.

## Introduction

Over the past 20 years, coronaviruses have posed severe threats to public health. In 2003, severe acute respiratory syndrome coronavirus (SARS-CoV-1) emerged in humans after being transferred from an animal reservoir and infected over 8000 people with a fatality rate of ~10% fatality rate (*Ksiazek et al., 2003*; *Florindo et al., 2020*). Middle East respiratory syndrome coronavirus (MERS-CoV) has infected over 1700 people with a fatality rate of ~36% since 2012 (*Zaki et al., 2012*). In late December 2019, a novel coronavirus, called severe acute respiratory syndrome coronavirus 2 (SARS-CoV-2), was identified as the cause of an outbreak of a new respiratory illness named COVID-19. SARS-CoV-2 has caused more than 4 million deaths to date. Considerable efforts have been made to understand its molecule mechanism.

Coronaviruses are large, enveloped, positive-stranded RNA viruses belonging to the coronaviridae family, which comprises four genera: alpha-coronaviruses, beta-coronaviruses, gamma-coronaviruses, and delta-coronaviruses (*Zumla et al., 2016*). SARS-CoV-2, SARS-CoV-1, and MERS-CoV, which infect

mammalians (*Wu et al., 2020*), are beta-coronaviruses. Envelope-anchored spike proteins are capable of mediating coronavirus entry into host cells by first binding to a specific host receptor and then fusing the viral and host membranes (*Wu et al., 2020*; *Yuan et al., 2017*). The coronavirus spike protein, a class I fusion protein, is synthesized as a precursor single polypeptide chain consisting of three segments: a large ectodomain, a single-pass transmembrane anchor, and a short intracellular tail (*Figure 1A*). After interacting with the host receptor, the spike protein is cleaved into an amino-terminal subunit (S1) and a carboxyl-terminal subunit (S2) by host furin-like proteases (*Yuan et al., 2017*; *Li, 2016*; *Lan et al., 2020*). The receptor-binding domain (RBD) located in the C-terminal region of the S1 subunit (S1 CTD) is responsible for recognizing and binding the host receptor and is critical in determining the cell tropism, host range, and zoonotic transmission of coronaviruses (*Wu et al., 2020*; *Li, 2016*). The S2 subunit contains a hydrophobic fusion loop for membrane fusion. Cryo-EM studies have illuminated the prefusion and postfusion structures of the SARS-CoV-1 and SARS-CoV-2 spike proteins, implying that coronaviruses undergo conformational changes during infection. The spike protein forms a clove-shaped homotrimer with three S1 heads and a trimeric S2 stalk. Structural comparisons indicated that spike protein utilizes the CTD1 (N-terminal domain in CTD) as the RBD, which changes from the 'down' conformation to the 'up' conformation and then converts from the inactivated state to the activated state to allow for receptor binding, and possibly also to initiate subsequent conformational changes of the S2 subunits to mediate membrane fusion (*Yuan et al., 2017*; *Li, 2016*; *Lan et al., 2020*; *Kirchdoerfer et al., 2018*). The binding of spike RBD to SARS-CoV-2 and ACE2 is the first step in viral entry into the host cell. Thus, the majority of vaccines and neutralizing antibodies that are under development target this region.

Recently, several variants with increased transmissibility have been found. The alpha variant (B.1.1.7 lineage) was first detected in the United Kingdom in September 2020, and another variant (B.1.351 lineage, beta) was first detected in October 2020 in the Republic of South Africa (*Rahimi and Talebi Bezmin Abadi, 2021*; *Greaney et al., 2021*; *CNBC, 2021*). Both of these mutants carry an N501Y mutation, and the B.1.351 lineage has two additional mutations (K417N and E484K) within the RBD region (*Figure 1B,C*; *Leung et al., 2021*; *Tu et al., 2021*). Recent data suggest that the FDA-authorized mRNA vaccines continued to induce a high level of neutralization against the B.1.1.7 variant, but a lower level against B.1.351 variants. Several researchers have assessed the neutralization potency of numerous antibodies against the two new variants. Their data suggest that some neutralizing antibodies in phase II/III clinical trials were not able to retain their neutralizing capability against the B.1.351 variant. As these mutations are within the RBD region, an understanding of the mechanism that allows the new variants to bind to the ACE2 receptor is of great value.

The structure of the RBD–ACE2 complex showed that extensive interactions are formed between RBD and ACE2 (*Lu et al., 2020*; *Bauer et al., 2020*). To understand the potential role of the RBD mutations in binding to ACE2, we combined a cell-surface-binding assay, a kinetics study, a single-molecule biophysical method, and steered molecular dynamics (SMD) simulations to study the interaction of RBD mutants and ACE2 (*Figure 1D,E*). Our results reveal the molecular mechanism that underlies the increased transmissibility of two SARS-CoV-2 variants by identifying the key mutation N501Y. This information could be valuable for the development of further vaccines and neutralizing antibodies against mutant forms of the SARS-CoV-2 virus.

## Results

### Cell-surface binding of RBD to ACE2

To elucidate the interaction between the RBD of SARS-CoV-2 variants and ACE2, we performed a cell-surface-binding assay. ACE2 with a mCherry fused at the C-terminus was transfected into HEK293 cells. Confocal microscopic images subsequently showed that ACE2 was located mainly in the cell membrane and endoplasmic reticulum (*Figure 1F*). ACE2–mCherry-positive cells were stained with AlexaFluor488 labeled-RBD, and the overlay images showed the co-localization of RBD and ACE2 on the cell surface (*Figure 1F*), validating their interaction.

To measure the binding affinity of RBD from SARS-CoV-2 for ACE2 on the surface of cells, saturation binding was performed using fluorescence flow cytometry by titration of Alexa488-RBD without washing the cells, which yielded a $K_d$ of about 50 nM (*Figure 1G*). We also performed a

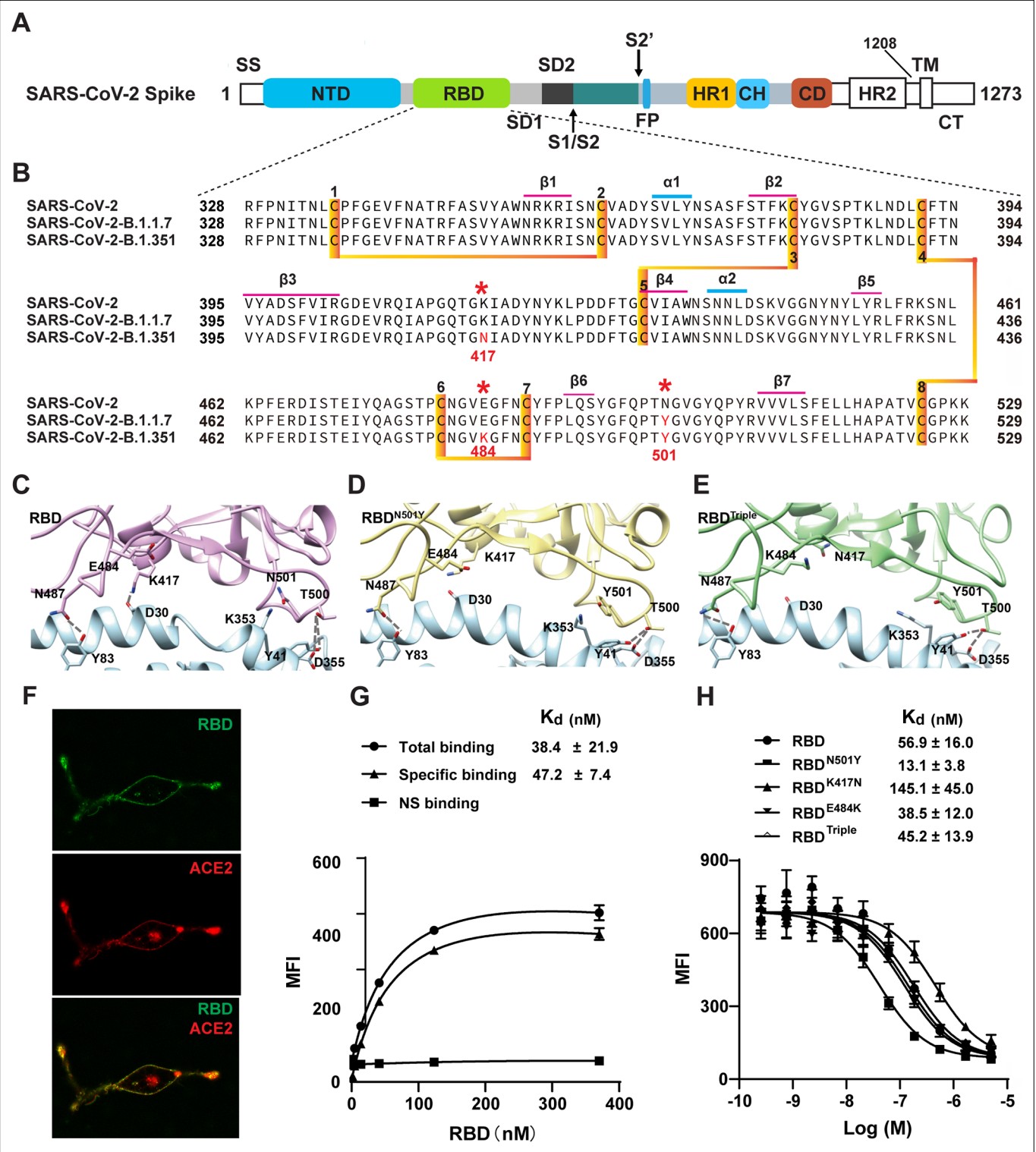

**Figure 1.** Two SARS-CoV-2 variants bind to ACE2 with higher affinity. (**A**) Domain architecture of the SARS-CoV-2 spike monomer. NTD, N-terminal domain; RBD, receptor-binding domain; SD1, subdomain 1; SD2, subdomain 2; FP, fusion peptide; HR1, heptad repeat 1; CH, central helix; CD, connector domain; HR2, heptad repeat 2; TM, transmembrane region; CT, C-terminal. (**B**) Sequence alignment of RBD from SARS-CoV-2, B.1.1.7, and B.1.351 variant spike proteins. The N501Y, K417N, and E484K mutations are highlighted in red with a *. Cysteines forming disulfide bonds are marked in orange. (**C–E**) The interface of ACE2 (cyan) in complex with spike RBD from SARS-CoV-2 (violet), B.1.1.7 lineage (yellow), and B.1.351 lineage (green). Residues 501, 500, 417, 487, and 484 from the RBD and the mutant RBD, and the contacting residues from ACE2 (Y41, K353, D355, D30, and Y83) are shown in sticks. Hydrogen bonds are shown in dash lines. (**F**) Representative images of ACE2–mCherry (red) HEK293 cells stained with 100 nM

*Figure 1 continued on next page*

*Figure 1 continued*

AlexaFluor488-labeled RBD (green). (**G**) Saturated binding of AlexaFluor488-labeled RBD to cell-surface ACE2. NS, non-specific. (**H**) Series-diluted RBD and RBD mutants were incubated with ACE2-expressing cells in the presence of AlexaFluor488-labeled RBD protein (100 nM). Concentrations used for unlabeled RBD and RBD mutants were from 5 µM to 0.25 nM with threefold dilution. $K_d$ values were calculated using the Cheng–Prusoff equation.

The online version of this article includes the following figure supplement(s) for figure 1:

**Source data 1.** Source cell-surface-binding data used for *Figure 1G,H*.

competition-binding assay by titrating unlabeled RBD to compete with 100 nM Alexa488-RBD, which showed a similar affinity of 50 nM (*Figure 1H*).

## N501Y mutation slowed the dissociation of the RBD from the ACE2 receptor

To determine the role of the receptor mutations, we first compared the ability of all of the RBD mutants to bind to the cell surface with that of wild-type RBD using a competition binding assay (*Figure 1H*). The N501Y mutation from the B.1.1.7 variant showed a fourfold greater affinity than wild-type RBD for the cell surface. The mutation resulted in a slightly weaker or similar affinity to cell-surface ACE2, whereas the N501Y, K417N, E484K triple mutation resulted in an affinity similar to that of the wild type RBD. These results demonstrated that N501Y is the key residue change that increases binding affinity.

To further understand the changes in kinetics that result from the RBD mutations, we performed surface plasmon resonance (SPR) on the immobilized RBD or RBD mutants with ACE2 as an analyte (*Figure 2A–C*, thin black lines). Compared to RBD, both RBD$^{N501Y}$ and RBD$^{Triple}$ showed a 10-fold increase in affinity, which resulted from a significantly lower off-rate and a slightly higher on-rate (*Figure 2*). Two other amino acid mutations (K417N and E484K) had less impact on ACE2 binding, as verified by two single-point mutants (*Figure 2—figure supplement 1*). This result again emphasized

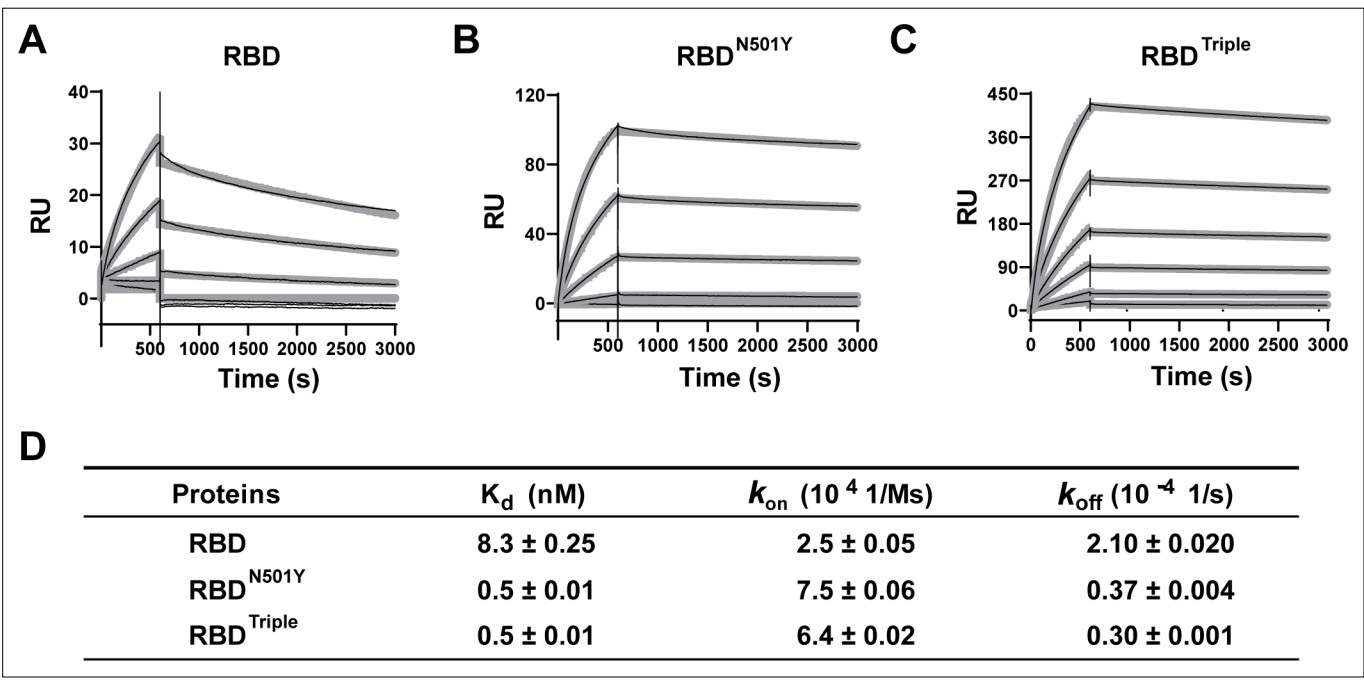

| Proteins | $K_d$ (nM) | $k_{on}$ ($10^4$ 1/Ms) | $k_{off}$ ($10^{-4}$ 1/s) |
|---|---|---|---|
| RBD | 8.3 ± 0.25 | 2.5 ± 0.05 | 2.10 ± 0.020 |
| RBD$^{N501Y}$ | 0.5 ± 0.01 | 7.5 ± 0.06 | 0.37 ± 0.004 |
| RBD$^{Triple}$ | 0.5 ± 0.01 | 6.4 ± 0.02 | 0.30 ± 0.001 |

**Figure 2.** Kinetics of the binding of the receptor-binding domain (RBD) and of RBD mutants to the ACE2 protein. (**A–C**) Surface plasmon resonance (SPR) sensorgrams (thin black lines) with fits (thick gray lines). ACE2 protein concentrations of 50, 20, 10, 5, 2, and 1 nM were used. Values were fitted to the 1:1 binding model. (**D**) $K_d$ and kinetic rates are shown as fit ± fitting error.

The online version of this article includes the following figure supplement(s) for figure 2:

**Figure supplement 1.** Kinetics of RBD$^{K417N}$ and RBD$^{E484K}$ binding to ACE2 protein.

**Source data 1.** Source data describing the kinetics of each receptor-binding domain (RBD) bound to ACE2 protein used for *Figure 2* and *Figure 2—figure supplement 1*.

the role of N501Y, rather than the other two mutations, in increasing binding affinity by slowing the rate of dissociation from the ACE2 receptor.

## AFM showed a higher binding probability and binding strength for the two variants containing N501Y

In addition to classic ensemble measurements, we used atomic force microscopy (AFM)-based single-molecule force spectroscopy (SMFS) to measure the strength of binding between the three different RBDs and ACE2 on a living cell directly (*Alsteens et al., 2017a*; *Hinterdorfer and Dufrêne, 2006*). AFM-SMFS is a powerful single-molecule nanotechnology that can be used like optical and magnetic tweezers to manipulate a single molecule or several molecules mechanically (*Jobst et al., 2015*; *Walder et al., 2017*; *Alonso-Caballero et al., 2021*; *Löf et al., 2019*; *Huang et al., 2019*; *Zhang et al., 2019*; *Xing et al., 2020*). It has been widely used to study protein mechanics and protein–protein interactions, including the interaction between the spike proteins of viruses and living cells (*Kim et al., 2010*; *Cuellar-Camacho et al., 2020*; *Yang et al., 2020*; *Sieben et al., 2012*; *Alsteens et al., 2017b*; *Yu et al., 2019*; *Ma et al., 2017*). A previous AFM experiment identified the binding events between the wild-type RBD and human ACE2 transfected on A549 cells, obtaining their binding probability and unbinding force/kinetics (*Yang et al., 2020*). In our work, a single RBD is site-specifically immobilized to a peptide-coated AFM tip via an enzymatic ligation (*Figure 3A*, step 1) (*Deng et al., 2019*; *Ott et al., 2018*; *Yuan et al., 2019*; *Popa et al., 2016*). An N-terminal GL sequence is present in the peptide, and a C-terminal NGL is added to the three RBDs. These two sequences can be recognized and ligated by protein ligase *Oa*AEP1 into a peptide bond linkage, and the RBD is attached to the tip for AFM measurement (*Deng et al., 2019*). Then, we used the ACE2–mCherry-transfected HEK293 cells as the target cell, which is immobilized on a Petri dish coated with poly-D-lysine. With the help of fluorescence, we targeted the transfected cell for measurement (*Figure 3B*).

Upon moving the AFM tip towards the cell, the RBD contacts the cell and binds to the ACE2 on the surface (steps 2 and 3). Then, the tip retracts at a constant velocity and pulls the complex apart by breaking all the interactions, leading to a force-extension curve with a force peak corresponding to the unbinding of the RBD–ACE2 complex (steps 3 and 4, *Figure 3A*; *Rief et al., 1997*). If the RBD does not bind to the ACE2 receptor, a featureless curve will be observed (*Figure 3C*, curve 1). Finally, the tip moves to another spot (65 nm away) on the cell and repeats the cycle for tens of hundreds of times, leading to a force map of the unbinding force distribution of RBD over the cell surface (*Figure 3E*; *Hinterdorfer et al., 1996*; *Müller et al., 2009*).

Because previous ensemble measurements showed that the N501Y mutation contributes most to the higher binding affinity of the RBD, we mainly focus on RBD$^{N501Y}$, RBD$^{Triple}$, and wild-type RBD for AFM-SMFS studies and comparisons. For example, 2815 data points on force-extension curves (using a pulling speed of 5 μm/s) have been obtained by probing ACE2-transfected cells with a RBD$^{N501Y}$-functionalized AFM tip . As indicated by an unbinding force of >20 pN, 14%  of the events involved a specific interaction between RBD$^{N501Y}$ and ACE2 (*Figure 3D* curve 4, *Figure 3—figure supplement 1*). The same experiments and analysis were performed for RBD (3.3%) and RBD$^{Triple}$ (11.2%). The interaction between RBD and untransfected normal HEK293 cells (1.7%) was also measured as a control (*Figure 3—figure supplement 2a*). The mean unbinding forces for RBD, RBD$^{N501Y}$, and RBD$^{Triple}$ were 49 ± 11 pN (n = 349), 57 ± 18 pN (n = 394), and 56 ± 12 pN (n = 312), respectively (*Figure 3C*). Both RBD mutants showed a higher binding probability and unbinding force than the wild type, while the properties of the two variants are similar to each other (*Figure 3C,D*, *Figure 3—figure supplement 2B-D*). The absolute difference in force value between the two mutations and the wild type is small (~10 pN),  but  this is still a difference of ~15%  difference due to their low unbinding force (~50 pN). Finally, we also performed AFM measurements on RBD$^{K417N}$ (40 ± 11 pN, n = 894) and RBD$^{E484K}$ (41 ± 11 pN, n = 606) (*Figure 3—figure supplement 3*). These mutant RBDs showed weaker unbinding forces than wild type RBD, in agreement with previous ensemble measurements and verifying our single-molecule results.

Moreover, AFM-SMFS can also be used to obtain the unbinding kinetics, which further support our previous conclusions. According to the Bell–Evans model, the force that is externally applied by AFM lowers the unbinding activation energy (*Merkel et al., 1999*; *Yu et al., 2017*; *Zheng and Li, 2011*; *Garcia-Manyes and Beedle, 2017*). Thus, the binding strength of the ligand–receptor bond (i.e. interaction) is proportional to the logarithm of the loading rate, which describes the effect of the

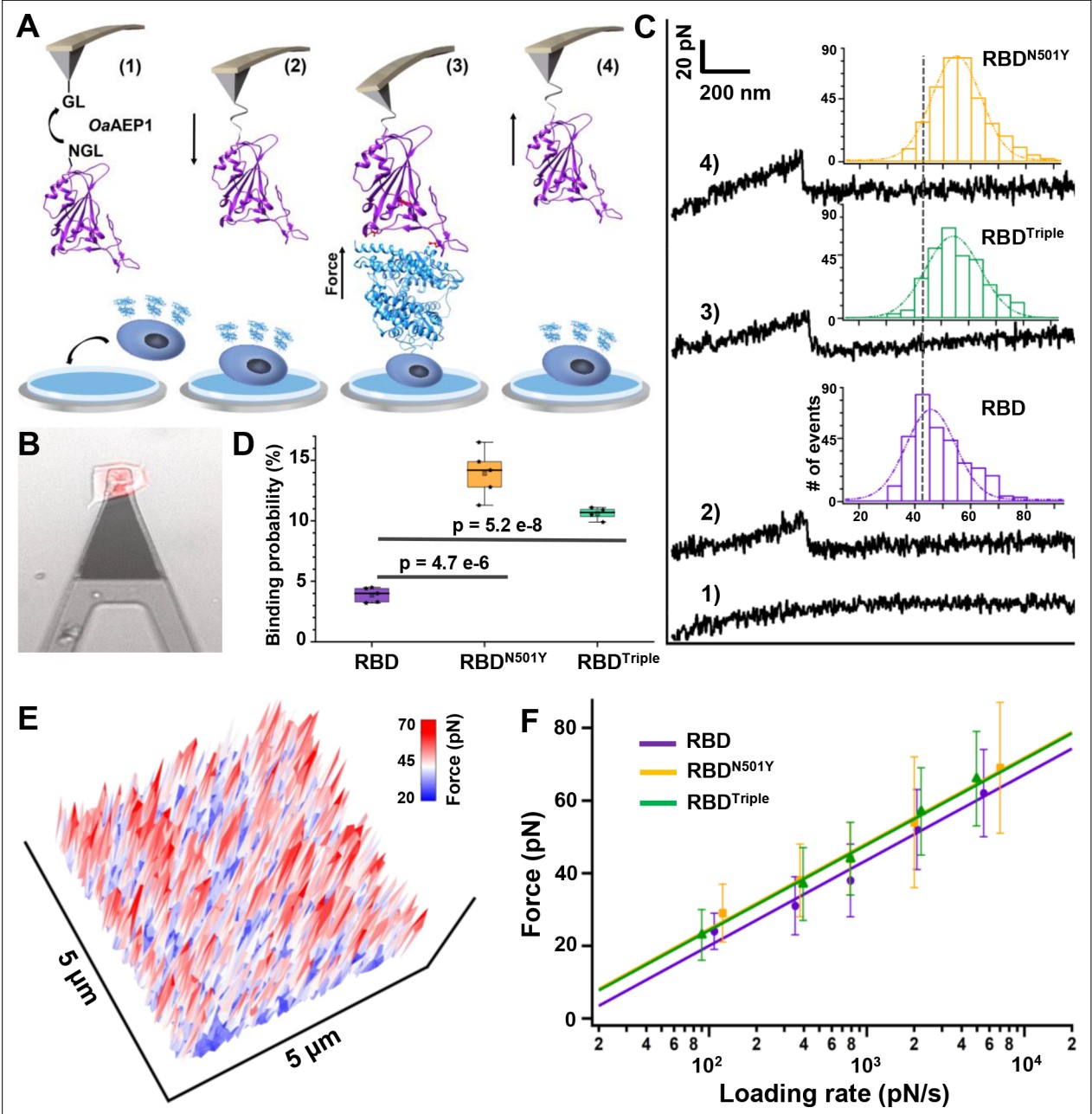

**Figure 3.** Atomic force microscopy-based single-molecule force spectroscopy (AFM-SMFS) experiment to quantify the strength of binding between the receptor-binding domains (RBDs) and ACE2 in living cells. (**A**) Schematic of the AFM-SMFS measurement process showing how the interaction is quantified. RBD with an N-terminal NGL recognition sequence is immobilized on a GL-coated AFM tip by the ligase *Oa*AEP1, which recognizes the two sequences and ligates them to form a peptide bond (1). As the AFM tip approaches the target cell (2), RBD binds to ACE2 (3). Then the tip retracts, and the complex dissociates, leading to an unbinding force peak (4). (**B**) The reddish ACE2–mCherry-transfected HEK293 cell is measured under the AFM tip by an inverted fluorescent microscope. (**C**) Representative force-extension curves show no binding event (curve 1) and specific binding events between RBD–ACE2 complexes with an unbinding force peak (curves 2–4). In the force histograms (inset), RBD$^{N501Y}$ and RBD$^{Triple}$ show higher unbinding forces (57 pN and 56 pN) than the RBD (49 pN). (**D**) Box plot of the specific binding probabilities between the three RBDs and target cells from AFM experiments, indicating a higher binding probability for the two mutants under five different velocities. The box indicates the 25th and 75th percentiles. (**E**) 3D AFM force mapping of the cell surface showed the unbinding force distribution. (**F**) A plot of loading rate against the most probable unbinding forces from the complexes showed a linear relationship. The data are fitted to the Bell–Evans model to extract the off-rate.

The online version of this article includes the following figure supplement(s) for figure 3:

**Source data 1.** Source data for the histograms of unbinding force for different RBD–ACE2 complexes used in *Figure 3C* and *Figure 3—figure supplement 3*.

*Figure 3 continued on next page*

*Figure 3 continued*

**Source data 2.** Binding probabilities for different RBD–ACE2 complexes used for *Figure 3D*.

**Source data 3.** Force mapping results for the different complexes used for *Figure 3E* and *Figure 3—figure supplement 4*.

**Source data 4.** Loading rates for different RBD–ACE2 complexes used for *Figure 3F*.

**Source data 5.** The spring constant (k) for all 15 cantilevers.

**Figure supplement 1.** Representative force-extension curves for different RBD–ACE2 complexes involving (**A**) RBD, (**B**) RBD$^{N501Y}$, (**C**) RBD$^{Triple}$, (**D**) RBD$^{K417N}$, or (**E**) RBD$^{E484K}$.

**Figure supplement 2.** The force mapping results for the three different complexes in a 2 × 2 μm area: (**A**) blank on untransfected normal HEK293 cell, (**B**) RBD, (**C**) RBD$^{N501Y}$, and (**D**) RBD$^{Triple}$.

**Figure supplement 3.** Histograms of unbinding force for RBD$^{K417N}$–ACE2 (**A**) and RBD$^{E484K}$–ACE2 (**B**) under a pulling speed of 5 μm/s.

**Figure supplement 4.** Plots of raw unbinding force against the loading rate of the three different complexes: (**A**) RBD, (**B**) RBD$^{N501Y}$, and (**C**) RBD$^{Triple}$.

force applied on the bond over time. Thus, we pulled the RBD–ACE2 complexes apart at different velocities, and plotted the relationship between the unbinding forces and loading rate (*Figure 3F*, *Figure 3—figure supplement 4*). From the fit (SI), we can estimate the bond dissociation rate ($k_{off}$) and the length scale of the energy barrier ($\Delta x_\beta$) (*Hickman et al., 2017*). Similarly, the $k_{off}$ of the two RBD mutants are close to each other ($0.030 \pm 0.017$ s$^{-1}$ and $0.035 \pm 0.024$ s$^{-1}$) but slower than that of the wild-type RBD ($0.075 \pm 0.048$ s$^{-1}$).

## SMD simulations revealed a higher unbinding force for the complexes due to additional π-π and cation-π interactions resulting from the N501Y mutation

To explore the possible molecular mechanism of RBD–ACE2 complex dissociation under force, we performed SMD simulations to visualize the unbinding process that took place during the AFM study (*Figure 4 and Figure 4—video 1*, *Figure 4—video 2*, *Figure 4—video 3*; *Dong et al., 2017*; *Milles et al., 2018*; *Tian et al., 2020*; *Mei et al., 2020*). In the wild-type RBD–ACE2 interaction, T500 forms two hydrogen bonds with Y41 and D355 from ACE2; K417 from RBD forms a salt bridge with D30 from ACE2; and N487 forms one hydrogen bond with Y83 (*Figure 4A*, snapshot 1). The broken forms of these interactions, except the interaction between N487 and Y83, showed the highest rupture force during the simulations and are regarded as the most critical step for the dissociation of the complex . This force was defined as the rupture or disassociation force for the complex and was simulated as 427 ± 58 pN (n = 20, *Figure 4A*, *Figure 4—figure supplement 1A*). Then, the interaction between N487 and Y83 was broken, leading to the complete dissociation of the complex (*Figure 4A*, snapshot 2).

In the RBD$^{N501Y}$–ACE2 complex, Y501 forms an additional π-π interaction with Y41 and an additional π-cation interaction with K353 (*Figure 4B*, snapshot 1). The rupture of these additional interactions as well as the interactions that are also present in the wild-type complex (snapshot 1) led to the highest rupture force during the simulation. An elevated unbinding force of 499 ± 67 pN (n = 20) was obtained for the complex between RBD$^{N501Y}$ and ACE2 compared to that for the complex between wild-type RBD and ACE2 (p=8.29 e$^{-4}$, *Figure 4B*, *Figure 4—figure supplement 1B*). Similarly, SMD simulations revealed that RBD$^{Triple}$ might also have a stronger contact than wild-type RBD with ACE2. The unbinding force was 521 ± 65 pN (p=2.33 e$^{-5}$, *Figure 4C*, *Figure 4—figure supplement 1C*).

In addition to the complex dissociation force, we also measured the distance between these key residues (Y(N)501–Y41 and Y(N)501–K353) during the simulations (see Materials and methods for details). For the wild-type RBD–ACE2 complex, the distances between residue N501 and residues Y41 (*Figure 4D*) and K353 (*Figure 4E*) increased significantly at a shorter extension along the pulling pathway (purple line) at the distance of 2.2 ± 0.7 nm (n = 20), indicating no interaction between these residues in the RBD–ACE2 complex. For the two variants (*Figure 4D,E*), the distances showed a smaller increase at an extension of 2.9 ± 0.8 nm (p=5.38 e$^{-3}$, orange line, n = 20) and 3.3 ± 0.8 nm (p=6.39 e$^{-5}$, green line, n = 20), respectively. Consequently, the SMD simulations further demonstrate that the RBD$^{N501Y}$–ACE2 and RBD$^{Triple}$–ACE2 complexes have a higher unbinding force than wild-type RBD–ACE2 complexes, and that the additional π-π and cation-π interactions that result from the N501Y mutation may provide the molecular mechanism that underlies this result.

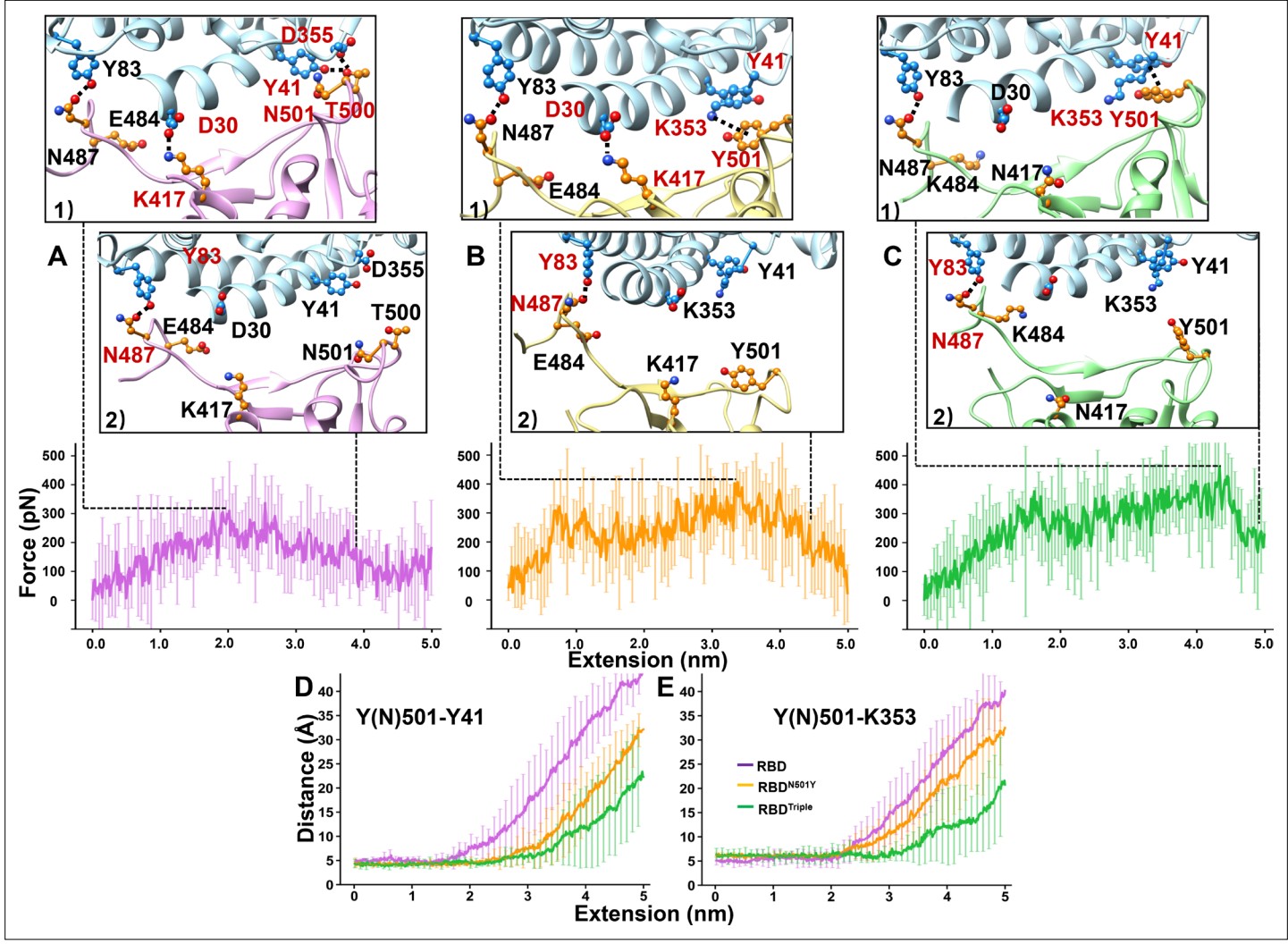

**Figure 4.** SMD simulations of RBD–ACE2 complex dissociation. (**A–C**) Force-extension traces of RBD–ACE2 (violet), RBD^N501Y–ACE2 (orange), and RBD^Triple–ACE2 (green) complexes pulled apart at 5 Å/ns. The curves represent the average results from 20 simulations, with the standard deviations represented by vertical lines. In the ribbon diagrams, ACE2 is colored in cyan. Snapshots 1 and 2 represent the changes that occur as the RBDs are dissociated from ACE2 sequentially. The residues that are involved in the interaction between RBDs and ACE2 are labeled and depicted in sphere models, and the residue is colored in red if the interaction in which it participates is ruptured in the snapshot. (**D, E**) The distances between the Y41 (left) and K353 (right) residues of the ACE2 receptor and the Y(N)501 and residue of RBD (colored in purple), RBD^N501Y (orange) or RBD^Triple (green) as the extension elongates.

The online version of this article includes the following video and figure supplement(s) for figure 4:

**Source data 1.** Rupture forces for the three complexes from 20 SMD simulations.

**Source data 2.** Distances between key residues in the RBDs–ACE2 complexes used for *Figure 4D,E*.

**Figure supplement 1.** SMD simulations of the dissociation of the different RBD–ACE2 complexes involving (**A**) RBD, (**B**) RBD^N501Y or (**C**) RBD^Triple.

**Figure 4—video 1.** Representative SMD simulation of the RBD (violet)–ACE2 complex at a constant velocity of 5.0 Å/ns.
https://elifesciences.org/articles/69091/figures#fig4video1

**Figure 4—video 2.** Representative SMD simulation of the RBD^N501Y(orange)–ACE2 complex at a constant velocity of 5.0 Å/ns.
https://elifesciences.org/articles/69091/figures#fig4video2

**Figure 4—video 3.** Representative SMD simulation of the RBD^Triple (green)–ACE2 complex at a constant velocity of 5.0 Å/ns.
https://elifesciences.org/articles/69091/figures#fig4video3

## Discussion

Coronaviruses are large, enveloped, positive-stranded RNA viruses that have a remarkable mutation rate that allows them to evolve in a way that affects their transmission. In this study, we combined cell-surface-binding assays, mechanical manipulation by AFM-SMFS, and molecular dynamics simulations to understand the behavior of key mutations recently detected in B.1.1.7 and B.1.351 variants that affect RBD binding. All of these methodologies suggested that, of the three mutations examined in this work, the N501Y mutation in the RBD has the most significant role in binding and dissociation from the ACE2 receptor . The cell surface-binding assay showed that, when compared wild-type RBD, RBD$^{N501Y}$ had greater binding affinity for ACE2. SPR and AFM-SMFS measurements consistently showed that RBD$^{N501Y}$ had increased $k_{on}$ and binding probability and decreased $k_{off}$. It is noted that SPR was performed on the complex between RBD and isolated ACE2 protein, whereas the other two measurements were performed on the complex between RBD and ACE2 on living cells. The more complex cell surface of living cells, where other proteins or receptors may interact with RBD in a non-specific and weaker way, may account for the relatively weak results observed at the cellular level. Indeed, the AFM unbinding results indicated a fraction of non-specific interactions between RBDs and the surface of untransfected normal cells (~5% for a RBD monomer on HEK293 cells studied here, ~10% for a S1/RBD homotrimer on A549 cells) (*Alsteens et al., 2017a*). Previous AFM results on wild-type RBD showed a higher specific binding probability than that observed in this study (~20% vs. ~15%), possibly resulting from different RBD construct and immobilization methods and the use of different host (A549) cells (*Yang et al., 2020*; *Deng et al., 2019*; *Cao et al., 2020*). Nevertheless, the specific unbinding forces detected for the RBD–ACE2 complex (~50 pN) in the two experiments are similar. Moreover, the two mutants showed higher binding probability and unbinding force, as well as lower off-rate, under the same conditions in our studies.

One concern was whether ACE2 was pulled out of the cell membrane during our AFM experiments. First, the unbinding force that was measured (~50 pN) is much smaller than the typical force needed to pull a membrane-bound protein out of a cell membrane (~100 pN). Also, if this had happened, the isolated ACE2 would stick to the RBD-functionalized coverslip and would block further measurement. In our AFM experiment, the pick-up ratio was mostly consistent with the measurements. Finally, previous work on purified ACE2 protein and wild-type RBD showed a rupture force of ~50 pN (*Yang et al., 2020*; *Cao et al., 2020*). Thus, we believe that the ACE2 was not pulled out of the cell membrane during our experiments.

There was only a small difference in complex stability between RBD$^{N501Y}$ and RBD$^{Triple}$, where two more mutations are present in RBD$^{Triple}$. Indeed, the results from flow cytometry showed that E484K contributed less to the interaction increment than N501Y, whereas K417N even decreased the interaction. The effects of the E484K and K417N mutations may cancel each other out, and N501Y is the dominant site in affecting the interaction. Thus, the combination of all three of these mutations in RBD$^{Triple}$ lead to an effect that was similar to that of RBD$^{N501Y}$. Indeed, a similar $k_{off}$ value is obtained for RBD$^{N501Y}$ and RBD$^{Triple}$ from AFM-SMFS measurement.

Finally, the SMD simulations provide valuable information on the enhanced interaction between the RBD mutants and ACE2. It is noted that only a modest effect from the N501Y mutation was detected in the experimental data, such as the 10 pN difference in complex dissociation force between the wild-type and variants detected by AFM measurement. Consequently, we performed a range of different experimental biophysical methods to measure the kinetics and confirm these differences. Thus, the results of SMD simulations are essential parts of our work, revealing that additional interactions are present for the N501Y mutant RBD and are responsible for the increased rupture force. Owing to computational limitations, we chose 0.5 m/s as the pulling speed for the SMD simulations, which is a normal speed for simulations. Nevertheless, this speed is much higher than the experimental speed (5 μm/s), although it does not affect the molecular insights provided by the simulations. As confirmed by other simulation studies performed at various speeds (*Kim et al., 2021*; *Pavlova et al., 2021*; *Han et al., 2021*), our simulations revealed that additional π-π and cation-π interactions resulting from the N501Y mutation lead to the higher rupture force. In addition, a higher pulling speed typically leads to a higher rupture or dissociation force. Thus, the simulation also confirmed rupture force trend for the three complexes. Nevertheless, all of these methods have their own advantages and limitations. The experiment provides quantitative results under physiological conditions, whereas the simulations provide an underlying molecular mechanism that rarely obtained by experimentation (*Kim et al.,*

*2021*; *Zhou et al., 2019*). Thus, we believe that the combination of experimental and simulation techniques used here is necessary and provides a complete picture of this important problem.

Another SARS-CoV-2 variant, P.1 lineage (gamma), was identified in January 2021 in Brazil. Studies found that, like the two variants studied in this work, it may also affect the ability of antibodies to recognize and neutralize the virus (*Hodcroft et al., 2021*). Interestingly, three key mutations (N501Y, E484K, and K417T) are found in the RBD of the P.1 variant. Consequently, we believe that N501 is a critical mutation that affects the transmission of COVID-19 by strengthening the interaction between RBD and ACE2. The strong interaction of the N501Y mutant RBD leads to the tighter binding of SARS-CoV-2 to the host cell, allowing complete membrane fusion or the internalization of the receptor with the virus. Thus, we speculate that the enhanced affinity of SARS-CoV-2 variants for host cells may contribute to the observed increase in infectivity by lowering the effective virion concentration required for cell entry. A modest change in affinity could cause a significant increase in the infection rate. Indeed, the alpha variant of SARS-CoV-2, which contains the N501Y mutation, had become the most common lineage in the United States by June 2020. It is noted that several other SARS-CoV-2 variants, such as delta, epsilon, and kappa, have been found very recently during the revision of our work (*Kim et al., 2021*). Some of these variants have very high transmission rates and have been proposed to be more dangerous. Although they do not contain the N501Y mutation, other mutations in the RBD might increase the binding of the virus to the human receptor ACE2, leading to a similar effect. As an RNA virus, SARS-CoV-2 is evolving rapidly. Hundreds of millions of human beings have been affected, and the virus is still spreading rapidly, so future dangerous mutations causing even higher transmission to humans or other creatures are very possible. Thus, further surveillance, diagnosis, evaluation, and treatment of mutated SARS-CoV-2 strains are necessary.

Nowadays, most vaccines are designed on the basis of the RBD or spike protein. Neutralizing antibodies from these vaccines mainly target the RBD to weaken its binding to ACE2. Here, we found that a RBD with the N501Y mutation has a 5–10 times higher affinity than wild-type RBD for ACE2. Additional mutations within the receptor-binding site (K417N and E484K) changed the amino acid sequence of the epitope and may contribute to the escape from antibody binding. Thus, a higher vaccine-induced antibody titer or neutralizing antibodies of higher affinity are needed to compete for the RBD$^{N501Y}$–ACE2 interaction (*Chen et al., 2021*; *Focosi and Maggi, 2021*). This will make the current vaccine less effective against SARS-CoV-2 variants that contain the N501Y mutation.

# Materials and methods
## Protein expression and purification

The genes were ordered from GenScript Inc The RBD construct contains the SAS-COV-2 spike protein (residues 319–591), followed by a GGGGS linker and an 8XHis tag in a pcDNA3.4 modified vector (SI). Its mutants, including RBD$^{N501Y}$, RBD$^{K417N}$, RBD$^{E484K}$, and RBD$^{Triple}$ (N501Y, K417N, E484K), were generated using the QuikChange kit. Their sequences were all verified by direct DNA sequencing (GENERAL BIOL). A C-terminal NGL was added to the RBD for use in the AFM-SMFS experiment. The human ACE2 construct contains the ACE2 extracellular domain (residues 19–740) and an Fc region of IgG1 at the C-terminus.

All RBD and human ACE2 proteins were expressed in Expi293 cells with OPM-293 CD05 serum-free medium (OPM Biosciences). For protein purification of RBD with His-tag, culture supernatant was passed through a Ni-NTA affinity column (Qiagen), and Fc-tag ACE2 protein was purified using a protein affinity A column (Qiagen). Proteins were further purified by gel filtration (Superdex 200 Increase 10/30 GL, GE Healthcare). The full-length ACE2 construct contains the ACE2 protein (residues 1–805), followed by a GGSGGGGS linker and a mCherry tag in a pcDNA3.4 modified vector. The ACE2 expression cell line was constructed by transient transfection of HEK293 cells and used for flow cytometry and AFM in this study.

OaAEP1(C247A) is cysteine 247 to alanine mutant of asparaginyl endoprotease 1 from *Oldenlandia affinis*, abbreviated as *Oa*AEP1 here (*Yang et al., 2017*; *Shi et al., 2021*). ELP is the elastin-like polypeptide (*Ott et al., 2017*). *Oa*AEP1 and ELP were overexpressed in BL21(DE3) *Escherichia coli*. Luria-Bertani (LB) medium and agar plates (Sangon Biotech) were used for the culture of *E. coli*. Protein concentrations were routinely determined by Nanodrop 2000. Details of the protein expression and

purification protocols can be found in the literature (*Deng et al., 2019*; *Yang et al., 2017*; *Ott et al., 2017*).

## Confocal microscopy

A confocal microscope was used to detect binding between SARS-CoV-2 RBD and the ACE2 receptor on the cell surface. Briefly, ACE2–mCherry cells were seeded into a poly-D-lysine precoated confocal dish and stained with AlexaFluor488-labeled-RBD (100 nM) for 30 min at room temperature (RT). After three washes, the samples were imaged on a confocal microscope (Zeiss LSM 710), and the images were prepared using ZEN software.

## Cell-surface binding by flow cytometry

For saturation binding, wide-type RBD protein was labeled with AlexaFluor488 NHS Ester (Yeasen) according to the manufacturer's instructions. ACE2 expression cells (mCherry positive) and control HEK293 cells (mCherry negative) were resuspended in PBS buffer and incubated with AlexaFluor488 labeled-RBD at 4°C for 1 hr, and then subjected to flow cytometry without washing. Mean fluorescence intensity (MFI) of AlexaFluor488 was reported for total binding (mCherry positive) and non-specific binding (mCherry negative). $K_d$ value was calculated from the saturation-binding curve.

For the competitive binding, ACE2 expression cells were resuspended in PBS buffer and incubated with 100 nM AlexaFluor488-labeled RBD in the presence of competitors (0–5000 nM of unlabeled-RBD and RBD mutants). The mixture was allowed to equilibrate for 1 hr before flow cytometry (ThermoFisher Attune NxT) without washing. The binding of RBD mutants was reported by changes in the MFI of AlexaFluor488. The decrease in the MFI value was directly proportional to the increase in the concentration of competitors. Competition curves were fitted using nonlinear regression with top and bottom values shared. The $IC_{50}$ value was converted to an absolute inhibition constant $K_d$ using the Cheng–Prusoff equation (*Newton et al., 2008*).

## Surface plasma resonance

SPR studies were performed using Biacore T200 (GE Healthcare). Purified RBD and RBD mutants were amine-immobilized on the CM5 chips. Purified ACE2 protein was injected at 20 µL/min in 0.15 M NaCl, 20 mM HEPES, pH 7.4. The surface was regenerated with a pulse of 25 mM HCl at the end of each cycle to restore resonance units to baseline. Kinetics analysis was performed with SPR evaluation software version 4.0.1 (GE Healthcare).

## AFM tip functionalization

First, the maleimide group for cysteine coupling was added onto the amino-functionalized AFM tip using the hetero-bifunctional crosslinker sulfosuccinimidyl 4-(N-maleimidomehthyl) cyclohexane-1-carboxylate (Sulfo-SMCC, Thermo Scientific) from the reaction between amino and -NHS. Then, the peptide GL-ELP$_{20}$-C was added to the maleimide. The long ELP$_{20}$ serves as a spacer that prevents non-specific interactions between the tip and the cell surface, and is a signature for the single-molecule event. Finally, target RBDs with the C-terminal NGL sequence were site-specifically linked to the tip by the ligase *Oa*AEP1, which recognized the N-terminal GL on the tip and the C-terminal NGL, forming a peptide bond (*Deng et al., 2019*; *Song et al., 2021*).

Specifically, we used the silicon nitride AFM cantilever (MLCT-BIO-DC, Bruker Corp.). The tip was coated with the amino group by amino-silanization (*Ebner et al., 2019*). Briefly, the cantilevers were immersed in 1.5% (v/v) APTES toluene solution for 1 hr at RT in the dark and rinsed with ethyl alcohol. After drying, they were baked at 80 °C for 15 min and then cooled down to RT. 200 µL of sulfo-SMCC (1 mg/mL) in dimethyl sulfoxide (DMSO) were added and incubated for 2 hr, while protected from light. The cantilevers were washed with absolute ethyl alcohol to remove residual sulfo-SMCC. Finally, each cantilever was reacted with more than 50 µL of GL-ELP$_{20}$-C and washed with 50 mL high-salt buffer (100 mM Tris, 1 M NaCl, pH 7.4), and then dried. The cantilevers were then ready for RBD immobilization.

To add RBD to the AFM tip, the GL-functionalized cantilevers were incubated with a 50 µL of a mixed solution of 60 µM RBD-NGL and 1 µM *Oa*AEP1 in the measurement buffer. The *Oa*AEP1-catalyzed coupling was performed in the measurement buffer (100 mM Tris, 100 mM NaCl, pH 7.4) at RT for ~30 min, forming a covalent NGL linkage between the AFM tip and the RBD.

## AFM-SMFS experiment

AFM (Nanowizard4, JPK) coupled to an inverted fluorescent microscope (Olympus IX73) was used to acquire correlative images and the force-extension curve. The AFM and the microscope were equipped with a cell culture chamber that allowed the temperature to be maintained at 37°C ± 1 °C. Fluorescence images were recorded using a water-immersion lens (×10, numerical aperture (NA) 0.3). Optical images were analyzed using ImageJ software.

The D tip of the MLCT-Bio-DC cantilever (Bruker) was used to probe the interaction between the RBD and ACE2 on the cell. Its accurate spring constant was determined by a thermally induced fluctuation method (*Hutter and Bechhoefer, 1993*). The peptide linker C-ELP$_{20}$-GL was used to functionalize the AFM tips as previously described (*Deng et al., 2019*). Typically, the tip contacted the cell surface for 400 ms under an indentation force of 450 pN, thereby ensuring a site-specific interaction between the RBD on the tip and ACE2 on a cell while minimizing the non-specific interaction. The sample was scanned using 32 × 32 pixels per line (1024 lines) and a sample number of 10,500. Then, moving the tip up vertically at a constant velocity (5 μm/s, if not specified), the complex ruptured. Then, the tip was moved to another location to repeat this cycle several thousands of times. As a result, a force-extension curve, possibly including the complex unbinding event, was obtained. AFM images and force-extension curves were analyzed using JPK data process analysis software.

## Bell–Evans model to extract kinetics

The dissociation of the RBD–ACE2 complex in the AFM experiment is a non-equilibrium process that can be modeled as an all-or-none two-state process, with force-dependent rate constant $k$(F). The rate constant can be described by the Bell–Evans model (*Merkel et al., 1999*; *Evans and Ritchie, 1997*):

$$k\left(F\right) = k_{off}exp\left(\frac{Fx_\beta}{k_bT}\right)$$

(1)

where $k$(F) is the complex dissociation rate constant under a stretching force of F, $k_{off}$ is the dissociation rate constant under zero force, and $\Delta x_\beta$ is the distance between the bonded state and the transition state of dissociation.

For the dynamic force spectroscopy measurements, the slope $a$ of the force-extension curves immediately (2–3 nm) before the rupture event was first determined to obtain the average loading rate ($r = av$, where $r = av$ is the velocity in the rupture event). All of the data were fitted with the Bell–Evans model (1), yielding the spontaneous rupture rate, and the distance from the bound state to the transition state with the following equation:

$$F = \frac{k_bT}{x_\beta}ln\left(\frac{x_\beta}{k_{off}k_bT}\right) + \frac{k_bT}{x_\beta}ln\left(r\right)$$

(2)

By performing the AFM unbinding experiment under five different pulling speeds, 0.5 μm/s, 1 μm/s, 3 μm/s, 5 μm/s, and 10 μm/s, the relationship between the most probable rupture force and loading rate can be obtained on a log scale, which is fitted by a linear line as *equation (2)*. Thus, the slope of this line can be used to calculate the $\Delta x_\beta$, which is the distance between the bonded state and the transitional unbonded state, and the y-intercept is used to calculate the $k_{off}$.

## SMD simulation for the dissociation of RBD–ACE2 complex

The structure model for RBD in complex with the receptor ACE2 was taken from the Protein Data Bank (PDB:6M0J). The RBD mutants RBD$^{N501Y}$ and RBD$^{Triple}$ have no structure available and thus were created using the CHARMM36 force field (*Best et al., 2012*). Each system underwent a similar equilibration and minimization procedure. The molecular dynamics simulations were set up with the QwikMD plug-in in VMD (*Humphrey et al., 1996*), and simulations were performed employing the NAMD molecular dynamics package (*Phillips et al., 2005*). The CHARMM36 force field was used in all simulations. Simulations were performed with explicit solvent using the CHARMM TIP3P (*Zheng and Li, 2011*) water model in the NpT ensemble. The temperature was maintained at 300.00 K using Langevin dynamics. The pressure was maintained at one atmosphere using a Nosé–Hoover Langevin piston (*Kim et al., 2019*). A distance cut-off of 12.0 Å was applied to short-range, non-bonded interactions, and 10.0 Å for the smothering functions. Long-range electrostatic interactions were treated using the particle-mesh Ewald method. The motion equations were integrated using the r-RESPA

(*Phillips et al., 2005*) multiple time-step scheme to update the short-range interactions every one step and long-range electrostatic interactions every two steps. A time step of integration of 2 fs was chosen for all simulations. Before the molecular dynamics (MD) simulations, all of the systems were submitted to an energy minimization protocol for 1000 steps. An MD simulation with position restraints in the protein backbone atoms was performed for 1 ns, with temperature ramping from 0 K to 300 K in the first 0.25 ns, which served to pre-equilibrate the system before the steered MD simulations. The RBD mutations were subjected to 100 ns of equilibrium MD to ensure conformational stability. All structures shown are from post-equilibration MD simulations.

To characterize the interaction between RBDs and ACE2, the SMD simulations with constant velocity stretching employed a pulling speed of 5.0 Å/ns and a harmonic constraint force of 7.0 kcal/mol/Å (*Florindo et al., 2020*) was applied for 10.0 ns. In this step, SMD were employed by harmonically restraining the position of the C-terminus of ACE2 and pulling on the C-terminus of the RBD (wildtype or mutant). Each system was run 20 times. Simulation force-extension traces were analyzed analogously to experimental data. For each simulation, the key step was determined as the interaction of the key residues in the RBDs and ACE2. Data were analyzed by the MD 1.9.3 program and its plug-ins. In VMD, the distance between the center of mass (COM) of the benzene ring of Y41 from the ACE2 and the COM of the benzene ring of Y501 (sidechain of N501) from the RBDs, as well as the distance between the hydrogen atom on the sidechain nitrogen-hydrogen bond of K353 from the ACE2 and the COM of the benzene ring of Y501 (sidechain of N501) from the RBDs were measured. p-values were determined by two-sample *t*-tests in Origin. The numerical calculations in this paper were performed using computing facilities in the High-Performance Computing Center (HPCC) of Nanjing University.

# Additional information

## Funding

| Funder | Grant reference number | Author |
| --- | --- | --- |
| National Key Research and Development Program of China | 2020YFA0509000 | Xianchi Dong |
| Fundamental Research Funds for the Central Universities | 14380205 | Peng Zheng |
| Natural Science Foundation of Jiangsu Province | BK20200058 | Peng Zheng |
| Natural Science Foundation of Jiangsu Province | BK20202004 | Peng Zheng |
| Natural Science Foundation of Jiangsu Province | BK20190275 | Bei Tong |
| National Natural Science Foundation of China | 21771103 | Peng Zheng |
| National Natural Science Foundation of China | 21977047 | Peng Zheng |
| Jiangsu Scientific and Technological Innovations Platform named Jiangsu Provincial Service Center for Antidiabetic Drug Screening. | | Bei Tong |
| National Natural Science Foundation of China | 3217110056 | Xianchi Dong |

| Funder | Grant reference number | Author |
|--------|------------------------|--------|

The funders had no role in study design, data collection and interpretation, or the decision to submit the work for publication.

## Author contributions

Fang Tian, Data curation, Formal analysis, Investigation, Methodology, Software, Writing – original draft; Bei Tong, Data curation, Formal analysis, Investigation, Project administration, Supervision, Writing – original draft; Liang Sun, Shengchao Shi, Zibin Wang, Data curation, Formal analysis; Bin Zheng, Formal analysis, Software; Xianchi Dong, Peng Zheng, Conceptualization, Funding acquisition, Project administration, Resources, Supervision, Writing – original draft

## Author ORCIDs

Fang Tian  http://orcid.org/0000-0002-4212-6328
Bei Tong  http://orcid.org/0000-0002-6863-6019
Peng Zheng  http://orcid.org/0000-0003-4792-6364

## Decision letter and Author response

Decision letter https://doi.org/10.7554/eLife.69091.sa1
Author response https://doi.org/10.7554/eLife.69091.sa2

# Additional files

## Supplementary files

• Transparent reporting form

## Data availability

All data generated or analysed during this study are included in the manuscript and supporting files. Source data files have provided for Figures 1-4.

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
