## [Decision Letter]

**Acceptance summary:**

This study finds that the receptor-binding domains (RBD) of variants containing the N501Y mutation exhibits higher affinity for ACE2, which provides a possible explanation for their higher transmission for SARS-CoV-2. While follow-up studies will be undoubtedly required to more thoroughly establish this hypothesis, we believe that this manuscript, which has shown vast improvements during this peer-review process, will be broad of interest to *eLife* readers.

**Decision letter after peer review:**

Thank you for submitting your article "COVID-19 N501Y Mutation of Spike Protein Strengthens the binding to its Receptor ACE2" for consideration by *eLife*. Your article has been reviewed by 2 peer reviewers, one of whom is a member of our Board of Reviewing Editors, and the evaluation has been overseen by José Faraldo-Gómez as the Senior Editor. The reviewers have opted to remain anonymous.

The reviewers have discussed their reviews with one another and with the Editors. Based on this discussion, we would like to invite you to submit a revised version of your manuscript. The Reviewing Editor has drafted this letter to help you prepare a revised submission.

Essential revisions:

1. A concern is regarding the AFM measurement is how do the authors know that the AFM pulling procedure is not pulling ACE2 out of the cell membrane?

2. The authors provide estimates of error in the off-rates measured using SPR (Figure 2d). However, they do not for the k_off_ measured using AFM-SM. Why is the error estimates neglected in the latter case? As presented, the differences in k_off_ derived from AFM-SM seem small. Without knowing the errors in the estimates, it isn't easy to gauge whether the observed differences are meaningful. The authors should therefore address this.

3. Can the authors please comment on why they did not carry out AFM-SM experiments on the K417N and E484K mutants?

4. The B.1.351 variant exhibits higher transmission and is less sensitive to existing vaccines than wild-type. The N501Y mutation might explain the increased transmission rate. The authors should comment on whether the additional mutations render the vaccines less neutralizing to B.1.351 variant in the Discussion section.

5. Moreover, the effects they have identified might be statistically significant, but they are modest. The authors must put forward an argument to explain why this difference is enough to drive the known differences in the SARS-CoV-2 variants.

6. In the abstract, the authors stated that "Molecular dynamics simulations of RBD-ACE2 complexes indicated that the N501Y introduced additional π-π and π-cation interaction for the higher 11 force/interaction." Unfortunately, though the authors observed higher maximum forces during the simulated unfolding of the Triple mutant, they did not perform a detailed contact analysis that links it to the additional π-π and π-cation interaction. Without such analysis, one cannot know why they observed higher forces for the mutant. The simulations have not yielded any more information beyond what is already evident from the AFM-SM experiments. Put differently, the current simulations and analyses have added little biophysical insights.

7. In the caption of Figure 4, the authors state that "Arrows indicate the key step during the dissociation of these complexes, of which structures are shown in (b-d), respectively." What do they mean by key step, how was it determined, and is it consistent over many independent trajectories? Related to the latter, the authors appear only to carry out three independent simulations per system. They should repeat the simulations at least 20 times and show that their observations are converged or consistent across the 20 trajectories.

[Editors' note: further revisions were suggested prior to acceptance, as described below.]

Thank you for resubmitting your work entitled "N501Y Mutation of Spike Protein in SARS-CoV-2 Strengthens its Binding to Receptor ACE2" for further consideration by *eLife*. Your revised article has been evaluated by Aaron Frank as Reviewing Editor and José Faraldo-Gómez as Senior Editor.

While we recognize the manuscript has improved, there remain substantial issues that must be resolved prior to publication – as outlined below. We urge you to address all of these issues as thoroughly and convincingly as possible before submitting the next revision of the manuscript.

In the abstract reference is made to "COVID-19 variants". The variants are of the virus, and so should be "SARS-CoV-2 variants".

Regarding Response 1: The transition to the added paragraph (Page 12; Line 305) is abrupt. The authors might consider something like:

"One concern was whether ACE2 was pulled out of the cell membrane during our AFM experiments. First, the unbinding force measured (~50 pN) is much smaller than the typical force needed to pull a membrane-bound protein out of the cell membrane…."

Regarding Response 2: It is stated that "In addition, as this reviewer pointed out later, it is a modest effect with a relatively small difference. That's another reason we performed a range of different methods to measure the kinetics and confirm this difference." This rationale should be explicitly articulated in the discussion of the manuscript. It will have the reader understand why multiple biophysical approaches were employed and the value-added by combining these methods.

Regarding Response 5: It is stated that "Enhanced affinity of SARS-CoV-2 variants contribute to the increased infectivity by lowering the effective virion concentration required for cell entry. Thus, a modest change in affinity will cause a significant arising in the infection rate." However, no direct or supporting evidence is present that supports this statement. Thus, it would be more appropriate if the authors should state this as an untested hypothesis. For example, "We speculate that enhanced affinity of SARS-CoV-2 variants may contribute to the increased infectivity by lowering the effective virion concentration required for cell entry. Thus, a modest change in affinity could cause a significant increase in the infection rate."

Regarding Response 6:

– Are the traces (4A-4E) from one simulation or an aggregate of 20 simulations? If from multiple simulations, we urge that you plot the error (or variation) in the force (4A-4C) and distance (4D and 4E) at each extension so that readers can get a sense of the uncertainty in the simulation data.

– Also, the statement that "SMD simulations revealed a higher unbinding force due to additional interactions for the complex with RBD mutant" needs to be revised to enhance clarity.

– Also, on Pg. 10, Line 247, it is written that "In the wild-type RBD-ACE2 interaction, T500 forms two hydrogen bonds with Y41 and D355 from ACE2. K417 from RBD forms a salt bridge with D30 from ACE2 (Figure 4A, Snapshot 1)" and on Pg. 252 that "In the RBDN501Y-ACE2 complex, Y501 forms an additional π-π interaction with Y41 and an additional π-cation interaction with K353(Figure 4B, Snapshot 1)." However, the referenced figure does not highlight these contacts. It will help the reader if close-up views of all the contacts that refer to in the text are included. These images should specifically show the contacts between the center of mass (COM) of benzene rings of Y41 from the ACE2 and the benzene ring of Y501 (for the π-π contact) and the distance between the hydrogen atom on the sidechain nitrogen-hydrogen bond of K353 from the ACE2 and COM of the benzene ring of Y501 (π-cation interaction).

– After reading the responses and the caption, it is not clear what Snapshot 1 and 2 are. Figure 4 caption states that "Snapshots (1) and (2) represent the difference of RBDs dissociated from ACE2 sequentially." What does "the difference of RBDs dissociated from ACE2 sequentially" mean? The caption and the references to Snapshot 1 and 2 in the main text should be revised to make it understandable to the general reader.

– In the representative SMD videos, we suggest that you render the specific contact π-π interaction between Y501 and Y41 and the rupture of the π-cation interaction between Y501 and K353. These contacts, between the center of mass (COM) of benzene rings of Y41 from the ACE2 and the benzene ring of Y501 (for the π-π contact) and the distance between the hydrogen atom on the sidechain nitrogen-hydrogen bond of K353 from the ACE2 and COM of the benzene ring of Y501 (π-cation interaction), can be displayed as dashed lines. The distances associated with these contacts at each frame should also be displayed. It will also help if the first frame in the video is annotated with labels highlighting which residue is Y501, Y41, and K353, and which contact is the contact π-π and π-cation. These modifications will enhance the information content of these animations.

– It remains unclear how the rupture force is determined. Please clarify, in the revised manuscript, what criteria were used to determine when a rupture has occurred.

The Discussion section lacks appropriate citations. It can also need to be further expanded to better place the results presented in the manuscript within the broader context of what is already known. Moreover, a careful and thoughtful discussion of the specific caveats associated with the experiments and simulation techniques employed currently is lacking but should be included in the revised manuscript.

---

## [Author Response]

Essential revisions:1. A concern is regarding the AFM measurement is how do the authors know that the AFM pulling procedure is not pulling ACE2 out of the cell membrane?

Yes, this will be a problem if it happened. Fortunately, we are confident that our measurements did not contain such events.

First, the rupture force between ACE2 and RBD (mutants) we measured is ~50 pN, which is smaller than a typical force (>100 pN) needed to pull a membrane-bound protein out of the membrane. Thus, ACE2, a three-time transmembrane protein measured here, is most likely not pulled out.

Also, if this happened, the isolated ACE2 will stick to the RBD-functionalized coverslip and block further measurement. In our AFM experiment, the pick-up ratio is consistent along with the measurement.

Finally, previous work on purified ACE2 protein and WT-RBD showed a rupture force of ~50 pN. Thus, we believe that the ACE2 was not pulled out of the cell membrane during our measurement. We have added this discussion in our revised manuscript. Please see Page 12, the last paragraph.

2. The authors provide estimates of error in the off-rates measured using SPR (Figure 2d). However, they do not for the k_off_ measured using AFM-SM. Why is the error estimates neglected in the latter case? As presented, the differences in k_off_ derived from AFM-SM seem small. Without knowing the errors in the estimates, it isn't easy to gauge whether the observed differences are meaningful. The authors should therefore address this.

As suggested, we have added the fitting error for the k_off_ obtained by our dynamic AFM-SMFS experiment: RBD (0.075±0.048 s^−1^), RBD^N501Y^ (0.035±0.024 s^−1^) and RBD^Triple^ (0.030±0.017 s^−1^). We have updated our results in the manuscript. Please see Page 8, from lines 26-27. This fitting error is also present in Figure 3—figure supplement 4.

In addition, as this reviewer pointed out later, it is a modest effect with a relatively small difference. That’s another reason we performed a range of different methods to measure the kinetics and confirm this difference.

3. Can the authors please comment on why they did not carry out AFM-SM experiments on the K417N and E484K mutants?

From the SPR and cell binding assay, we found that the N501Y and the triple mutant showed the most significant change while the effect of K417N and E484K mutant are weaker. To understand why and which mutation in RBD increases its binding to ACE2, we focused on the characterization of N501Y and the triple mutant by AFM-SMFS.

Thanks to the comment, we have now performed similar AFM experiments for K417N and E484K RBD mutations. As expected, they both showed a weaker unbinding force, which agrees with the previous SPR. It verifies our previous conclusion and completes our work.

We have added the explanation and these results on Page 8, from lines 16 to 18, in the main text and a new Figure 3—figure supplement 3 in SI.

4. The B.1.351 variant exhibits higher transmission and is less sensitive to existing vaccines than wild-type. The N501Y mutation might explain the increased transmission rate. The authors should comment on whether the additional mutations render the vaccines less neutralizing to B.1.351 variant in the Discussion section.

The neutralizing antibody induced by the vaccines will bind only to the original epitope. The additional mutations (K417N and E484K) within the receptor-binding site lead to the change in the amino acid sequence of the epitope, and may contribute to the escape from antibody binding. Thus, the vaccines are less efficient at neutralizing B.1.351 variant.

We have added this comment on Page 13, from lines 29 to 31.

5. Moreover, the effects they have identified might be statistically significant, but they are modest. The authors must put forward an argument to explain why this difference is enough to drive the known differences in the SARS-CoV-2 variants.

Enhanced affinity of SARS-CoV-2 variants contribute to the increased infectivity by lowering the effective virion concentration required for cell entry. Thus, a modest change in affinity will cause a significant arising in the infection rate.

We have added this comment on Page 13, from line 24 to line 26.

6. In the abstract, the authors stated that "Molecular dynamics simulations of RBD-ACE2 complexes indicated that the N501Y introduced additional π-π and π-cation interaction for the higher 11 force/interaction." Unfortunately, though the authors observed higher maximum forces during the simulated unfolding of the Triple mutant, they did not perform a detailed contact analysis that links it to the additional π-π and π-cation interaction. Without such analysis, one cannot know why they observed higher forces for the mutant. The simulations have not yielded any more information beyond what is already evident from the AFM-SM experiments. Put differently, the current simulations and analyses have added little biophysical insights.

As suggested, we have performed a detailed analysis of our SMD simulations results, focusing on the additional π-π and π-cation interaction from the N501Y mutation. The relationship of the distance between these key residues (Y(N)501-Y41 and Y(N)501-K353) vs. extension (simulation time) during the simulations was analyzed (Figure 4).

For the WT RBD-ACE2 complex, the distances between residue N501 from the RBD and residues Y41 and K353 from the ACE2 increased significantly at a shorter extension/earlier time along the pulling/unbinding pathway (purple line) at the distance of 2.2±0.7 nm (*n*=20), indicating no interaction between these residues in the WT RBD-ACE2 complex.

In the RBD^N501Y^-ACE2 complex, the distances between Y501-Y41 and Y501-K353 showed a relatively modest increase at a longer extension/later time (orange line), at an extension of 2.9±0.8 nm (*n*=20), compared to results of WT RBD-ACE2 complex. Similarly, the distances showed a smaller increase at an extension of 3.3±0.8 nm (green line, n=20). Consequently, it proves that the asparagine to tyrosine mutation (Y501) may form a π-π interaction with Y41 and a π-cation interaction with K353 on ACE2, leading to the stronger interaction for the two mutations. We have added these results on Page 11, the first paragraph of the main text and as two graphs as Figure 4D and Figure 4E.

7. In the caption of Figure 4, the authors state that "Arrows indicate the key step during the dissociation of these complexes, of which structures are shown in (b-d), respectively." What do they mean by key step, how was it determined, and is it consistent over many independent trajectories? Related to the latter, the authors appear only to carry out three independent simulations per system. They should repeat the simulations at least 20 times and show that their observations are converged or consistent across the 20 trajectories.

To highlight the key step as pointed out by this reviewer, we redesigned Figure 4 and added three sequential snapshot graphs for each simulation curve. Snapshot (1) is the previously stated “key step”, which is the rupture of the complex. And snapshot (2) is the complete dissociation state. For the two mutants, the arrow corresponds to the rupture of the additional π-π interaction between Y501 and Y41, and the rupture of the π-cation interaction between Y501 and K353. For the WT, the key step corresponds to the rupture of the two hydrogen bonds (T500-Y41, T500-D355) and the salt bridge (K417-D30)(Snapshot 2).

Moreover, we repeated the SMD simulations 20 times per system as suggested. The simulated rupture force of the complex is 427±58 pN for RBD, 499±67 pN for RBDN501Y and 521±65 for RBD^Triple^ (*n*=20). Representative simulated force-extension curves were shown in Figure 4—figure supplement 1. These results agree well with our previous results. Together with our previous contact analysis (response to comment 6), the simulations are consistent and further support our previous conclusion. And we appreciate these comments from this reviewer.

We have added/modified the discussion from line 25 to 36 on Page 10. The previous figure 4 has been redesigned with this new information. And new Figure 4—figure supplement 1 and Supplement table S1 are added.[Editors' note: further revisions were suggested prior to acceptance, as described below.]

In the abstract reference is made to "COVID-19 variants". The variants are of the virus, and so should be "SARS-CoV-2 variants".

We are sorry for this mistake and have corrected it as suggested.

Regarding Response 1: The transition to the added paragraph (Page 12; Line 305) is abrupt. The authors might consider something like:"One concern was whether ACE2 was pulled out of the cell membrane during our AFM experiments. First, the unbinding force measured (~50 pN) is much smaller than the typical force needed to pull a membrane-bound protein out of the cell membrane…."

We thank this suggestion and have modified the text as suggested. Please see page 13, line 3.

Regarding Response 2: It is stated that "In addition, as this reviewer pointed out later, it is a modest effect with a relatively small difference. That's another reason we performed a range of different methods to measure the kinetics and confirm this difference." This rationale should be explicitly articulated in the discussion of the manuscript. It will have the reader understand why multiple biophysical approaches were employed and the value-added by combining these methods.

Response: We have modified the text and explicitly state it in the discussion. Please see page 13, line 20.

Regarding Response 5: It is stated that "Enhanced affinity of SARS-CoV-2 variants contribute to the increased infectivity by lowering the effective virion concentration required for cell entry. Thus, a modest change in affinity will cause a significant arising in the infection rate." However, no direct or supporting evidence is present that supports this statement. Thus, it would be more appropriate if the authors should state this as an untested hypothesis. For example, "We speculate that enhanced affinity of SARS-CoV-2 variants may contribute to the increased infectivity by lowering the effective virion concentration required for cell entry. Thus, a modest change in affinity could cause a significant increase in the infection rate."

We agree with this comment and modified our description. Please see page 14, from line 9.

Regarding Response 6:– Are the traces (4A-4E) from one simulation or an aggregate of 20 simulations? If from multiple simulations, we urge that you plot the error (or variation) in the force (4A-4C) and distance (4D and 4E) at each extension so that readers can get a sense of the uncertainty in the simulation data.

The previous traces were individual representative curves from all simulations for each mutant. Thanks to the comment, we have now replaced them with curves drawn from the average result of the 20 times simulations, showing the standard deviation. And the caption in Figure 4 and the text have been changed accordingly.

– Also, the statement that "SMD simulations revealed a higher unbinding force due to additional interactions for the complex with RBD mutant" needs to be revised to enhance clarity.

We have modified the text as “SMD simulations revealed a higher unbinding force for the complex due to additional π-π and cation-π interactions from the N501Y mutation”.

– Also, on Pg. 10, Line 247, it is written that "In the wild-type RBD-ACE2 interaction, T500 forms two hydrogen bonds with Y41 and D355 from ACE2. K417 from RBD forms a salt bridge with D30 from ACE2 (Figure 4A, Snapshot 1)" and on Pg. 252 that "In the RBDN501Y-ACE2 complex, Y501 forms an additional π-π interaction with Y41 and an additional π-cation interaction with K353(Figure 4B, Snapshot 1)." However, the referenced figure does not highlight these contacts. It will help the reader if close-up views of all the contacts that refer to in the text are included. These images should specifically show the contacts between the center of mass (COM) of benzene rings of Y41 from the ACE2 and the benzene ring of Y501 (for the π-π contact) and the distance between the hydrogen atom on the sidechain nitrogen-hydrogen bond of K353 from the ACE2 and COM of the benzene ring of Y501 (π-cation interaction).

We have modified all the snapshots in Figure 4A-C by adding the dashed line to indicate all the contacts in a close-up view. And these residues that participate in the interactions are labeled as well.

– After reading the responses and the caption, it is not clear what Snapshot 1 and 2 are. Figure 4 caption states that "Snapshots (1) and (2) represent the difference of RBDs dissociated from ACE2 sequentially." What does "the difference of RBDs dissociated from ACE2 sequentially" mean? The caption and the references to Snapshot 1 and 2 in the main text should be revised to make it understandable to the general reader.

We are sorry for this unclear description. There are two major rupture steps of the interactions during SMD simulation. The first step involves the rupture of most interactions (except the Y83-N487 interaction) and leads to the highest rupture force during SMD simulations, which is the key/critical step upon the complex dissociation. Then, only the Y83-N487 interaction remains, which is broke at the second step, leading to the complete dissociation of the complex. Therefore, we have modified the caption and references to snapshot 1 and 2 in the main text. Please see Page 10, from line 26.

– In the representative SMD videos, we suggest that you render the specific contact π-π interaction between Y501 and Y41 and the rupture of the π-cation interaction between Y501 and K353. These contacts, between the center of mass (COM) of benzene rings of Y41 from the ACE2 and the benzene ring of Y501 (for the π-π contact) and the distance between the hydrogen atom on the sidechain nitrogen-hydrogen bond of K353 from the ACE2 and COM of the benzene ring of Y501 (π-cation interaction), can be displayed as dashed lines. The distances associated with these contacts at each frame should also be displayed. It will also help if the first frame in the video is annotated with labels highlighting which residue is Y501, Y41, and K353, and which contact is the contact π-π and π-cation. These modifications will enhance the information content of these animations.

We appreciate these detailed instructions. We have modified all the three videos as suggested.

– It remains unclear how the rupture force is determined. Please clarify, in the revised manuscript, what criteria were used to determine when a rupture has occurred.

The rupture force was the highest force detected during the SMD simulation, corresponding to Snapshot 1 in Figure 4A-C. We have clarified it in the main text. Please see Page 10, line 30.

The Discussion section lacks appropriate citations. It can also need to be further expanded to better place the results presented in the manuscript within the broader context of what is already known. Moreover, a careful and thoughtful discussion of the specific caveats associated with the experiments and simulation techniques employed currently is lacking but should be included in the revised manuscript.

As suggested, we have added seven references in the Discussion section, including very recent simulation work and antibody therapy for the new SARS-COV-2 variants (as follows). And a discussion is added. Please see Page 14, from line 12.

50. S Kim, Y Liu, Z Lei et al.,(2021) Differential Interactions Between Human ACE2 and Spike RBD of SARS-CoV-2 Variants of Concern. *bioRxiv*: 2021.2007.2023.453598.

51. A Pavlova, Z Zhang, A Acharya et al.,(2021) Machine Learning Reveals the Critical Interactions for SARS-CoV‑2 Spike Protein Binding to ACE2. *J. Phys. Chem. Lett.* 12: 5494-5502.

52. Y Han, Z Wang, Z Wei, I Schapiro, J Li (2021) Binding affinity and mechanisms of SARS-CoV-2 variants. *Comput. Struct. Biotechnol. J.* 19: 4184-4191.

53. W Zhou, G Fiorin, C Anselmi et al., (2019) Large-scale state-dependent membrane remodeling by a transporter protein. *eLife* 8: e50576.

54. E B Hodcroft, M Zuber, S Nadeau et al.,（2021）Spread of a SARS-CoV-2 variant through Europe in the summer of 2020. *Nature* 595: 707-712.

55. R E Chen, E S Winkler, J B Case et al., (2021) in vivo monoclonal antibody efficacy against SARS-CoV-2 variant strains. *Nature* 596: 103-108.

56. D Focosi, F Maggi (2021) Neutralising antibody escape of SARS‐CoV‐2 spike protein: Risk assessment for antibody‐based Covid‐19 therapeutics and vaccines. *Rev. Med. Virol.*

Finally, a discussion about the pros and cons of the experimental and simulation techniques we used has been added . Please see Page 13, from line 30.